# Low-Frequency *PPM1D* Gene Mutations Affect Treatment Response to CD19-Targeted CAR T-Cell Therapy in Large B-Cell Lymphoma

Katja Seipel [1,2,*] , Michèle Frey [2] , Henning Nilius [3] , Dilara Akhoundova [1,2] , Yara Banz [4] , Ulrike Bacher [5] and Thomas Pabst [2,*]

1 Department for Biomedical Research (DBMR), University of Bern, 3008 Bern, Switzerland; dilara.akhoundovasanoyan@insel.ch
2 Department of Medical Oncology, University Hospital Bern, 3010 Bern, Switzerland; michele.frey@students.unibe.ch
3 Department of Clinical Chemistry, University of Bern, 3010 Bern, Switzerland; henning.nilius@insel.ch
4 Institute of Tissue Medicine and Pathology (IGMP), University of Bern, 3010 Bern, Switzerland; yara.banz@unibe.ch
5 Department of Hematology, University Hospital Bern, 3010 Bern, Switzerland; veraulrike.bacher@insel.ch
* Correspondence: katja.seipel@unibe.ch (K.S.); thomas.pabst@insel.ch (T.P.);
Tel.: +41-31-6320934 (K.S.); +41-31-632-0378 (T.P.); Fax: +41-31-632-3410 (T.P.)

**Abstract:** Chimeric antigen receptor T (CAR T)-cell therapy has become a standard treatment option for patients with relapsed or refractory diffuse large B-cell lymphoma (r/r DLBCL). Mutations in the *PPM1D* gene, a frequent driver alteration in clonal hematopoiesis (CH), lead to a gain of function of PPM1D/Wip1 phosphatase, impairing p53-dependent G1 checkpoint and promoting cell proliferation. The presence of *PPM1D* mutations has been correlated with reduced response to standard chemotherapy in lymphoma patients. In this study, we analyzed the impact of low-frequency *PPM1D* mutations on the safety and efficacy of CD19-targeted CAR T-cell therapy in a cohort of 85 r/r DLBCL patients. In this cohort, the prevalence of *PPM1D* gene mutations was 20% with a mean variant allele frequency (VAF) of 0.052 and a median VAF of 0.036. CAR T-induced cytokine release syndrome (CRS) and immune effector cell-associated neuro-toxicities (ICANS) occurred at similar frequencies in patients with and without *PPM1D* mutations. Clinical outcomes were globally worse in the *PPM1D* mutated (PPM1Dmut) vs. *PPM1D* wild type (PPM1Dwt) subset. While the prevalent treatment outcome within the PPM1Dwt subgroup was complete remission (56%), the majority of patients within the PPM1Dmut subgroup had only partial remission (60%). Median progression-free survival (PFS) was 3 vs. 12 months ($p = 0.07$) and median overall survival (OS) was 5 vs. 37 months ($p = 0.004$) for the PPM1Dmut and PPM1Dwt cohort, respectively. Our data suggest that the occurrence of *PPM1D* mutations in the context of CH may predict worse outcomes after CD19-targeted CAR T-cell therapy in patients with r/r DLBCL.

**Keywords:** Diffuse Large B-cell Lymphoma (DLBCL); Chimeric Antigen Receptor T-cell (CAR T); Clonal Hematopoiesis (CH); Protein Phosphatase Mg/Mn-dependent 1D (PPM1D); Wild-type p53-induced phosphatase 1 (Wip1); Next-Generation Sequencing (NGS)





## 1. Introduction

Diffuse large B-cell lymphoma (DLBCL) is a common type of high-grade, fast-growing non-Hodgkin lymphoma (NHL), with an annual incidence of 4–8 cases in 100,000 people [1]. For the past 15 years, the standard first-line therapy for DLBCL has been a combined chemo-immunotherapy with R-CHOP (rituximab, cyclophosphamide, doxorubicin, vincristine, and prednisone) [2]. Although the therapy is safe and highly effective, 30–40% of patients have refractory disease or relapse after R-CHOP therapy [3]. Salvage therapy followed by high-dose chemotherapy with autologous stem cell transplantation (ASCT) is the standard

approach in younger patients (below the age of 60), as well as in older patients without therapy-limiting comorbidities [4]. However, despite this intensive treatment, 40–70% of patients experience a second relapse after ASCT, and the long-term outcome of those who relapse after transplant is poor [5–7].

Chimeric antigen receptor (CAR) T-cell therapy is a new immunotherapeutic option in the management of advanced lymphoproliferative malignancies, such as acute lymphatic leukemia (ALL), mantle cell lymphoma, as well as DLBCL [8,9]. With CD19-targeted CAR T-cell therapy, durable responses were achieved in 43–54% of r/r DLBCL patients [10]. While CAR T-cell persistence in the peripheral blood was associated with longer progression-free survival [11], common germline variants of the target antigen CD19 may also affect treatment outcomes of CAR T-cell therapy [12]. Currently, two FMC63-based anti-CD19 CAR T-cell products, axicabtagene ciloleucel (axi-cel, Yescarta©) and tisagen-lecleucel (Kymriah©), have been approved by the European Medicines Agency (EMA) for patients who received at least two prior treatment lines [9,13]. Moreover, axi-cel has been approved and is the new standard second-line therapy for younger patients with primary refractory disease or with early relapse, i.e., within 12 months of completion of first-line therapy [14,15]. CAR T-cell infusion may induce relevant toxicities, including cytokine release syndrome (CRS) in more than 60%, and immune effector cell-associated neurotoxicity syndrome (ICANS) in more than 30% of the recipients, as well as prolonged cytopenias [3,8,15,16]. CRS and ICANS are managed according to the severity of symptoms [17]. Standard management includes tocilizumab, a humanized monoclonal antibody against the interleukin-6 receptor, for patients with CRS, as well as corticosteroids in case of ICANS and/or CRS not responsive to tocilizumab [18,19]. Real-world studies reported an overall response rate of 52–82% after CAR T-cell therapy, with complete response (CR) rates of 40–52% [9,20–24]. Clinical response rates were associated with DLBCL histologic subtypes, as well as expansion levels and duration of persistence of CAR T-cells [6,15,25–27]. Patients relapsing after CAR T-cell therapy may be treated with glofitamab, a CD20- and CD3-targeting bispecific antibody [28]. Alternatively, the CD79b-directed antibody-drug conjugate polatuzumab vedotin has been approved in combination with bendamustine and rituximab after two or more prior therapies [29].

Clonal hematopoiesis (CH) refers to an expansion of clonally derived hematopoietic cells, without evidence of an underlying hematologic malignancy, and with a minimal VAF of 0.02. Mutations in genes that confer a selective advantage to hematopoietic stem cells drive CH. Most relevant CH driver genes include *PPM1D*, *TP53*, *DNMT3A*, *ASXL1*, and *TET2* [30–33]. In the general population, 0.13–5% carry a mutation in the *PPM1D* gene across all age groups [33,34], with an increase in somatic mutations with age [35]. More than 30% of patients with lymphoma who had undergone ASCT were found to have CH and *PPM1D* was the most frequent driver alteration [32,36]. Patients with CH-related *PPM1D* gene mutations are five times more likely to carry mutations in other genes, predominantly *TP53* [32,34,37]. The *PPM1D* gene is located on chromosome 17q, including six exons encoding the protein phosphatase Mg2+/Mn2+-dependent 1D (PPM1D, Wip1), an enzyme which targets the tumor suppressor protein p53 and other proteins involved in the DNA damage response (DDR) [34]. Genotoxic impacts can activate the "guardian of the genome" p53 to induce cell cycle arrest, DDR, and/or apoptosis of the damaged cell. For cell-cycle progression, Wip1 is targeted for degradation by the anaphase-promoting complex (APC/C), a multifunctional ubiquitin-protein ligase [38]. *PPM1D* alterations are typically nonsense or frameshift mutations in exon 6, resulting in truncated protein products promoting cell proliferation [33,39,40]. The truncated PPM1D protein variants are able to dephosphorylate p53, but are not recognized by the APC/C complex, leading to stabilization and accumulation of oncogenic mutant protein. Amplification and/or overexpression of *PPM1D* has been described in a significant number of solid tumors, suggesting a role for PPM1D in carcinogenesis [41–44]. In addition, studies have shown that the truncated protein products can lead to a relative fitness of hematopoietic cells in the presence of chemotherapy [34,45–47].

In lymphoma patients, upon treatment with R-CHOP, expansion of *PPM1D* mutant clones has been reported and occurred more commonly than for other CH-related genes [31,33,34]. *PPM1D* mutants show a selective advantage in the context of certain classes of chemotherapy, particularly drugs causing DNA cross-linking, as well as pyrimidine analogs and radiotherapy. In contrast, microtubule inhibitors do not appear to confer such an advantage [33,45,48]. Cell stress during ASCT is not driving *PPM1D* expansion, as studies have shown a decrease in the VAF of *PPM1D* mutant clones following ASCT [45]. Gibson et al. reported an association of *PPM1D* mutations with inferior overall survival (OS) after ASCT (10-year OS of 20.8% vs. 39.9%; *p* = 0.02 by log-rank test) [32]. Additionally, Lackraj et al. reported a significant association between *PPM1D* mutations and poor OS in DLBCL patients in the ASCT setting (HR 2.42, 95% CI 1.18–4.97, *p* = 0.016) [36]. Furthermore, Saini et al. reported elevated neurotoxicities after axi-cel in DLBCL patients with CH mutations [30]. Higher toxicities were primarily associated with DTA mutations (*DNMT3A*, *TET2*, and *ASXL1* genes), but not with mutations in *PPM1D* or *TP53*.

In this retrospective study, we determined the prevalence of *PPM1D* gene mutations in r/r DLBCL patients and analyzed the impact of *PPM1D* mutations on the safety and efficacy of CD19-targeted CAR T-cell therapy. The presence of mutations in *PPM1D* exon 6 was assessed through NGS amplicon sequencing of genomic DNA extracted from peripheral blood mononuclear cells (PBMC). We analyzed correlations between *PPM1D* mutational status and clinical outcomes after CAR T-cell therapy including response rates, survival times, as well as CAR T-cell-related toxicities.

## 2. Materials and Methods

### 2.1. Patient Samples

We conducted a retrospective single-center study at the Inselspital, University Hospital Bern, Switzerland. The analyzed cohort comprised 85 patients diagnosed either de novo or secondarily with DLBCL, who underwent commercial CAR T-cell therapy between January 2019 and August 2022. All participants gave written informed consent for the usage of personal data for research purposes. All patients were followed up clinically at one, three, six months, and one year after the CAR T-cell infusion. Imaging with PET-CT was performed three and twelve months post CAR T-cell infusion. In addition, clinical and laboratory data related to the underlying disease, the CAR T-cell treatment, and survival endpoints were systematically collected.

### 2.2. NGS Amplicon Sequencing

Genomic DNA was extracted from mononuclear cells (PBMCs) isolated from the peripheral blood of 88 DLBCL patients collected before CAR T-cell infusion and 8 healthy donors. NGS amplicon sequencing and bioinformatics analysis were performed at Microsynth, Balgach, Switzerland. NGS amplicon sequencing included preparation of a Nextera two-step PCR library, sequencing on Illumina MiSeq, 2 × 300 bases using a MiSeq Reagent Kit v3 (600 cycle) and gene-specific primers covering exon 6 of the PPM1D gene (F: 5′-GAGGATCCATGGCCAAGGG-3′, R: 5′-TTCCAATTTTCTTCTGGCCCC-3′, product size: 505 bp). Bioinformatics analysis included trimming of locus-specific Illumina adapter primers, merging of the reads, mapping trimmed and merged reads to human reference chromosome 17 for variant calling and annotation, and to the PPM1D-selected region (chr17:60,662,994-60,663,549) for coverage analysis, and dereplication of trimmed and merged reads. The total demultiplexed reads which passed Illumina's chastity filter were 3,124,600, and the demultiplexed bases were 933,189,886, with a mean read length of 299 bp. The quality of the reads was checked in fastq format with FastQC (version 0.11.9). Raw reads shorter than 200 bases, with average Q-values below 24, or incorporating uncalled 'N' bases were filtered using the BBTools software suite (version 38.96). The quality assessment returned a mean Q of 32, with 88% in Q20 and 77% in Q30. Mapping software bwa (version 0.7.17-r1198) in combination with samtools (version 1.15.1) was used to map remaining reads to (selected regions of) human UCSC hg38 reference genome

downloaded from iGenomes. Coverage analysis was performed with bedtools (version 2.30.0). Variant calling was performed with LoFreq software (version 2.1.5). Consequences of called variants were annotated on the amino acid level using annotation of the mentioned reference genome. Disclaimer: DNA library construction, sequencing, and data analysis described in this section were performed at Microsynth AG (Balgach, Switzerland).

### 2.3. Clinical Data Analysis

Parameters investigated for their potential prognostic significance were patient age, transformed versus de novo DLBCL, international prognostic index (IPI), previous treatment lines, prior radiotherapy, the need for bridging therapy before CAR T-cell infusion, remission status at the time of CAR T treatment, number of complete or partial response prior to CAR T therapy, prior SCT, lactate dehydrogenase (LDH) before lympho-depleting therapy, use of Kymriah®, Yescarta®, Bristol Myers Squibb® (Celgene®) CAR T-cell products, the manifestation of a cytokine release syndrome (CRS), and/or an immune effector cell-associated neurotoxicity syndrome (ICANS). Peak IL-6 and C-reactive protein (CRP) serum levels were recorded during the treatment in order to evaluate the occurrence of CRS and infections [17]. We investigated the remission status at follow-up dates at 100 days, 6 months, and 1 year after CAR T-cell infusion. The progression-free survival (PFS) and OS were defined as the time from CAR T-cell infusion to disease progression, death, or last follow-up, respectively. PFS and OS were censored at the last follow-up on 18 August 2023, which was also used as the data cutoff. Survival curves (Kaplan–Meyer) and univariate statistical analyses were performed on GraphPad Prism version 10 (GraphPad Software, San Diego, CA, USA). The categorical variables were summarized as frequencies and percentages, and the continuous variables as medians and ranges.

## 3. Results

### 3.1. Prevalence of PPM1D Mutations in r/r DLBCL

NGS amplicon sequencing was performed to identify mutations in exon 6 of the *PPM1D* gene in peripheral blood mononuclear cells isolated from 88 patients with r/r DLBCL before infusion of CD19-targeted CAR T-cells. Only *PPM1D* mutations with a variant allele frequency (VAF) > 0.015 were included. We identified 19 low-frequency *PPM1D* mutations in 18 genomic DNA samples of 88 DLBCL patients (18/88; 20.5%), with 10 in-del (53%; 10/19), 5 nonsense, and 4 missense mutations (Figure 1, Table 1). The identified *PPM1D* exon 6 mutations resembled those previously described [30,32,46], with a majority of stop-gain changes resulting in truncated protein products (14/18, 78%). The VAF of *PPM1D* mutations called in the DLBCL DNA samples ranged from 0.015 to 0.217, with a mean VAF of 0.052, and a median VAF of 0.036. One variant with a non-conservative change (F543V) was called in the reference healthy donor samples with a median VAF of 0.027, and in two DLBCL samples at VAF of 0.071 and 0.078. One variant with a conservative change (D509N) may be a benign variant. However, all variants with stop-gain changes are translated into phosphatase protein variants which may not be effectively targeted for degradation by the cellular APC/C complex. Consequently, the function of the p53 tumor suppressor may be impaired in cells with stop-gain mutated *PPM1D* genes, which may induce cell survival and proliferation (Figure 2).

### 3.2. Baseline Clinical Characteristics of the DLBCL Patient Cohort

Of the 88 r/r DLBCL patients with NGS data, 85 were admitted to CAR T-cell infusion. Clinical characteristics of the 85 r/r DLBCL patients are summarized in Table 2. The median age at first diagnosis was 61 years (range: 34–79 years) with 60% de novo DLBCL and 39% transformed DLBCL. Among patients diagnosed with transformed DLBCL, the most frequent initial diagnosis was follicular lymphoma (FL; *n* = 23), followed by chronic lymphocytic leukemia (CLL; *n* = 4). Most patients had initial disease stage IV (39%) and initial prognostic indexes of 3 or 4 (81%) with similar proportions in the PPM1D subgroups. Most patients had received three or more prior lines of therapy (65%) with

equal proportions in the PPM1D subgroups. Additionally, 45% of the PPM1Dwt and 67% of the PPM1Dmut population had undergone stem cell transplantation (SCT). The number of complete remissions and partial responses to previous treatments was equally distributed in both subgroups.

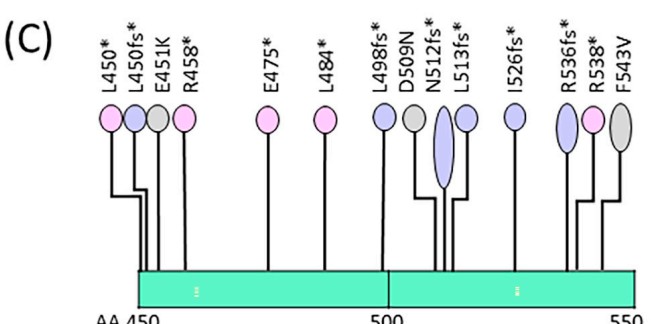

**Figure 1.** *PPM1D* gene and the encoded Wip1 protein phosphatase. (**A**) *PPM1D* gene coding region structure. (**B**) PPM1D protein phosphatase (Wip1) with central phosphatase domain and C-terminal degron region. (**C**) Degron region with lollipop plot indicating the positions of the 19 identified somatic mutations. NT: nucleotide; AA: amino acid. * (asterisk) = translation termination (stop) codon.

**Table 1.** *PPM1D* gene mutations detected in r/r DLBCL cohort.

| Classification | Locus Chr7 | VAF | NT Change | AA Change |
|---|---|---|---|---|
| indel | 60,663,077 | 0.079 | AT/A | L450fs * |
| nonsense | 60,663,083 | 0.023 | T/G | L450 * |
| missense | 60,663,085 | 0.036 | G/A | E451K |
| nonsense | 60,663,106 | 0.032 | C/T | R458 * |
| nonsense | 60,663,157 | 0.026 | G/T | E475 * |
| nonsense | 60,663,185 | 0.018 | T/A | L484 * |
| indel | 60,663,224 | 0.046 | del 17 | L498fs * |
| missense | 60,663,259 | 0.015 | G/A | D509N |
| indel | 60,663,262 | 0.217, 0.121, 0.017, 0.017 | C/CA | N512fs * |
| indel | 60,663,269 | 0.025 | AT/A | L513fs * |
| indel | 60,663,307 | 0.044 | GA/G | I526fs * |
| missense | 60,663,334 | 0.081, 0.078 | T/G | F543V |
| indel | 60,663,336 | 0.054 | TA/T | R536fs * |
| indel | 60,663,340 | 0.042 | AG/A | R536fs * |
| nonsense | 60,663,347 | 0.018 | T/G | L538 * |

Abbreviations: indel: insertion/deletion; VAF: Variant Allele Frequency; NT: Nucleotide; AA: Amino Acid. * (asterisk) = translation termination (stop) codon.

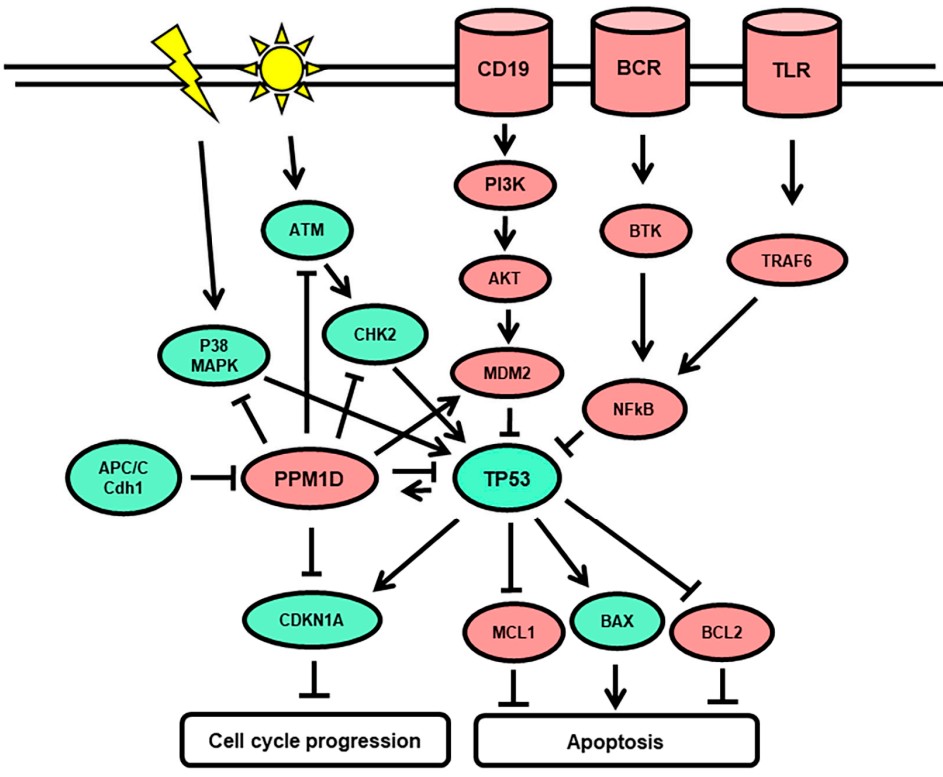

**Figure 2.** PPM1D phosphatase function in B-cells. Intracellular signaling initiated on the B-cell surface receptors CD19, BCR, and TLR leads to inhibition of the tumor suppressor p53 and to cell survival and proliferation. Radiation and genotoxic chemotherapy induce the activation of the tumor suppressor p53 leading to cell cycle arrest, DNA repair, and/or apoptosis. Protein phosphatase PPM1D (Wip1) and mutated/truncated PPM1D variants can dephosphorylate p53 and inhibit its activity. Anaphase-promoting complex/cyclosome (APC/C) targets the PPM1D protein and promotes its degradation by binding to the C-terminal degron region. Truncated PPM1D variants are not targeted by APC/C. Oncogenic functions are indicated in red, tumor suppressor functions in green, and DNA damage-inducing agents in yellow.

**Table 2.** Baseline clinical parameters of the r/r DLBCL cohort, univariate analysis.

|  | Cohort ($n$ = 85) | *PPM1D*$_{wt}$ ($n$ = 67) | *PPM1D*$_{mut}$ ($n$ = 18) | $p$-Value |
|---|---|---|---|---|
| Sex (female: male) | 34:51 | 26:40 | 7:11 | >0.99 |
| Median age at ID (range) | 61 (34–79) | 61 (34–79) | 61 (41–78) | 0.85 |
| Median age at CAR-T (range) | 66 (35–82) | 65 (35–82) | 69 (52–79) | 0.47 |
| DLBCL, de novo | 51 (60%) | 39 (58%) | 12 (67%) | 0.42 |
| DLBCL, transformed | 33 (39%) | 27 (40%) | 5 (28%) | |
| Initial Disease Stage | | | | |
| I | 3 (3%) | 2 (3%) | 1 (5%) | |
| II | 14 (16%) | 12 (18%) | 2 (11%) | |
| III | 14 (16%) | 9 (14%) | 5 (28%) | 0.73 |
| IV | 33 (39%) | 26 (39%) | 6 (33%) | |
| NR | 21 (25%) | 17 (25%) | 4 (22%) | |
| Prognostic Index (IPI) | | | | |
| 1 | 1 (1%) | 1 (2%) | 0 | |
| 2 | 8 (9%) | 6 (9%) | 2 (11%) | |
| 3 | 36 (42%) | 28 (42%) | 8 (44%) | 0.98 |
| 4 | 33 (39%) | 25 (37%) | 8 (44%) | |
| NR | 7 (8%) | 7 (10%) | 0 | |

**Table 2.** *Cont.*

| | Cohort (n = 85) | PPM1D$_{wt}$ (n = 67) | PPM1D$_{mut}$ (n = 18) | *p*-Value |
|---|---|---|---|---|
| Number of treatment lines prior to CAR-T therapy | | | | |
| 1 | 2 (2%) | 2 (3%) | 0 | >0.99 |
| 2 | 28 (33%) | 22 (33%) | 6 (33%) | |
| ≥3 | 55 (65%) | 43 (64%) | 12 (67%) | |
| Bridging chemotherapy | 35 (41%) | 27 (40%) | 8 (44%) | 0.99 |
| Bridging radiotherapy | 16 (19%) | 14 (21%) | 2 (11%) | 0.51 |
| Stem Cell Therapy (SCT) | 42 (49%) | 30 (45%) | 12 (67%) | |
| autologous | 41 | 30 | 11 | 0.31 |
| allogeneic | 1 | 0 | 1 | |
| Number of CR prior to CAR-T therapy | | | | |
| 0 | 35 (41%) | 28 (42%) | 6 (33%) | 0.59 |
| 1 | 35 (41%) | 27 (40%) | 8 (44%) | |
| 2 | 13 (15%) | 9 (14%) | 4 (22%) | |
| ≥3 | 1 (1%) | 1 (2%) | 0 | |
| Number of PR prior to CAR-T therapy | | | | |
| 0 | 32 (38%) | 24 (36%) | 8 (44%) | 0.77 |
| 1 | 40 (47%) | 31 (46%) | 8 (44%) | |
| 2 | 9 (11%) | 7 (11%) | 2 (11%) | |
| ≥3 | 3 (4%) | 3 (5%) | 0 | |

Significance of differences was calculated using chi-square test or Fisher's exact test; differences in median values were calculated using Mann–Whitney test. ID: initial diagnosis; DLBCL: diffuse large B-cell lymphoma; CAR T: chimeric antigen receptor T-cell; SCT: stem cell therapy.

### 3.3. Disease Features and CAR T-Cell Treatment

Disease status at the time of CAR T-cell infusion is presented in Table 3. The median time from first diagnosis to CAR T-cell infusion was 25 months for the PPM1Dwt and 33 months for the PPM1Dmut patient population, respectively. At the time of CAR T-cell treatment, the majority (57%) of patients in the PPM1Dwt subgroup had progressive disease (PD), while the majority (50%) of patients in the PPM1Dmut subgroup were in partial remission (PR). All patients received lympho-depleting chemotherapy for 3 days (day $-5$ to $-3$) with 300 mg/m$^2$ cyclophosphamide and 30 mg/m$^2$ fludarabine, with 2 days of washout prior to CAR T-cell infusion (day 0). Median pre-lympho-depletion LDH levels were lower in the PPM1Dwt patients compared to the PPM1Dmut patients (461 U/L vs. 602 U/L, $p = 0.075$). Three different CAR T-cell products were used: Kymriah® (Novartis, tisagenlecleucel; 62%), Yescarta® (Gilead, axicabtagene ciloleucel; 31%), and JCAR017-BCM-003 (Celgene®, 7%). The majority of the patients (56%) carried the CD19 gene polymorphism rs 2,904,880 with equal distribution in the PPM1D subgroups.

### 3.4. Clinical Outcome after CAR T-Cell Therapy

Table 4 illustrates the clinical outcome after CAR T-cell treatment. The majority of patients (80%) had cytokine release syndrome (CRS) after CAR T-cell infusion, mostly (62%) low grade 1. Immune effector cell-associated neuro-toxicity syndrome (ICANS) was detected in a third of the patients. While the majority of ICANS in the PPM1Dwt subgroup was low grade 1 or 2 (56%), ICANS in the PPM1Dmut subgroup tended to be high grade 3 or 4 (75%). Patients with CRS grade >= 2 were treated with tocilizumab, and patients with ICANS grade >= 2 were treated with steroids. The median duration of hospitalization was 21 vs. 23 days for the PPM1Dwt vs. PPM1Dmut population ($p = 0.34$). Inflammatory markers including C-reactive protein (CRP), ferritin, and IL-6 were elevated in the PPM1Dmut subgroup (CRP, 30 mg/L vs. 64 mg/L, $p = 0.105$; ferritin, 1209 μg/L vs.

1772 µg/L, *p* = 0.064; IL-6443 pg/mL vs. 559 pg/mL, *p* = 0.58). Treatment response differed between the two subgroups (*p* = 0.044). While the most frequent response in the PPM1Dwt cohort was CR (56%), the prevalent response in the PPM1Dmut subgroup was PR (60%). In the PPM1Dwt subgroup, relapses occurred in 35 patients (52%) after a median interval of 12 months and death in 29 patients (44%) after a median interval of 37 months after CAR T-cell infusion. Among the PPM1Dmut population, relapses occurred in 13 patients (72%) after a median interval of 3 months and death in 13 patients (72%) after a median interval of 5 months after CAR T-cell infusion. Survival times were longer in the PPM1Dwt compared to the PPM1Dmut population, with extended PFS (12 versus 3 months, *p* = 0.07) and significantly extended OS time (37 versus 5 months, *p* = 0.004) (Figure 3).

**Table 3.** Clinical characteristics and details of CAR T-cell treatments, univariate analysis.

| | Cohort (*n* = 85) | *PPM1D*$_{wt}$ (*n* = 67) | *PPM1D*$_{mut}$ (*n* = 18) | *p*-Value |
|---|---|---|---|---|
| CAR-T Product Kymriah®:Yescarta®:Celgene® | 53:26:6 | 40:22:5 | 13:4:1 | 0.4 |
| Target antigen variants CD19 +/−rs2904880 | 48:37 (56%) | 37:30 (55%) | 11:7 (61%) | |
| Stage at CAR-T cell infusion | | | | |
| CR | 5 (5.9%) | 5 (7.5%) | 0 | 0.79 |
| PR | 30 (35.3%) | 21 (31%) | 9 (50%) | 0.26 |
| SD | 4 (4.7%) | 3 (4.5%) | 1 (5.5%) | |
| PD | 46 (54.1%) | 38 (57%) | 8 (44%) | |
| Median interval ID to CAR-T cell infusion in months (range) | 26 (6–312) | 25 (6–312) | 33 (8–277) | 0.28 |
| Median LDH U/L prior CAR-T (range) | 490 (249–3949) | 461 (249–2355) | 602 (344–3949) | 0.075 |

Significance of differences was calculated using chi-square test or Fisher's exact test; differences in median values were calculated using Mann–Whitney test. CR: complete remission; PR: partial remission; SD: stable disease; PD: progressive disease; CAR T: chimeric antigen receptor T-cell; LDH: lactate dehydrogenase.

**Table 4.** Clinical outcome after CAR T-cell treatment, univariate analysis.

| | Total Cohort (*n* = 85) | *PPM1D*$_{wt}$ (*n* = 67) | *PPM1D*$_{mut}$ (*n* = 18) | *p*-Value |
|---|---|---|---|---|
| CRS | 68 (80%) | 52 (77%) | 16 (89%) | 0.57 |
| grade 1 | 42 (62%) | 32 (62%) | 10 (62%) | |
| grade 2 | 22 (32%) | 17 (33%) | 5 (31%) | |
| grade 3 | 3 (4.4%) | 3 (5.8%) | 0 | |
| grade 4 | 1 (1.5%) | 0 | 1 (6%) | |
| ICANS | 31 (36%) | 23 (34%) | 8 (44%) | 0.19 |
| grade 1 | 9 (29%) | 7 (30%) | 2 (25%) | |
| grade 2 | 6 (19%) | 6 (26%) | 0 | |
| grade 3 | 11 (35%) | 7 (30%) | 4 (50%) | |
| grade 4 | 5 (16%) | 3 (13%) | 2 (25%) | |
| Median Peak CRP mg/L (range) | 41.5 (3–328) | 30 (3–323) | 64 (3–328) | 0.105 |
| Median Peak IL-6 pg/mL (range) | 556 (4–157,117) | 443 (4–157,117) | 559 (7–42,209) | 0.58 |
| Median Peak Ferritin µg/L (range) | 1265 (99–13,393) | 1209 (99–13,393) | 1772 (290–12,398) | 0.064 |
| Admissions to IMC/ICU | 16 | 11 | 5 | 0.34 |
| Hospitalization time in days (range) | 21 (14–68) | 21 (14–68) | 23 (18–41) | 0.34 |

**Table 4.** *Cont.*

| | Total Cohort (*n* = 85) | *PPM1D*<sub>wt</sub> (*n* = 67) | *PPM1D*<sub>mut</sub> (*n* = 18) | *p*-Value |
|---|---|---|---|---|
| Best remission status post CAR T-cell therapy | *n* = 79 | *n* = 64 | *n* = 15 | |
| CR | 41 (52%) | 36 (56%) | 5 (33%) | |
| PR | 26 (33%) | 17 (27%) | 9 (60%) | 0.044 |
| SD | 4 (5%) | 3 (5%) | 1 (7%) | |
| PD | 8 (10%) | 8 (13%) | 0 | |
| Median Survival time Progression free (PFS) | 12 | | | |

Significance of differences was calculated using chi-square test or Fisher's exact test; differences in median values were calculated with the Mann–Whitney test. CR: complete response; PR: partial response; SD: stable disease; CRP: C-reactive protein; IMC/ICU: intermediate care unit.

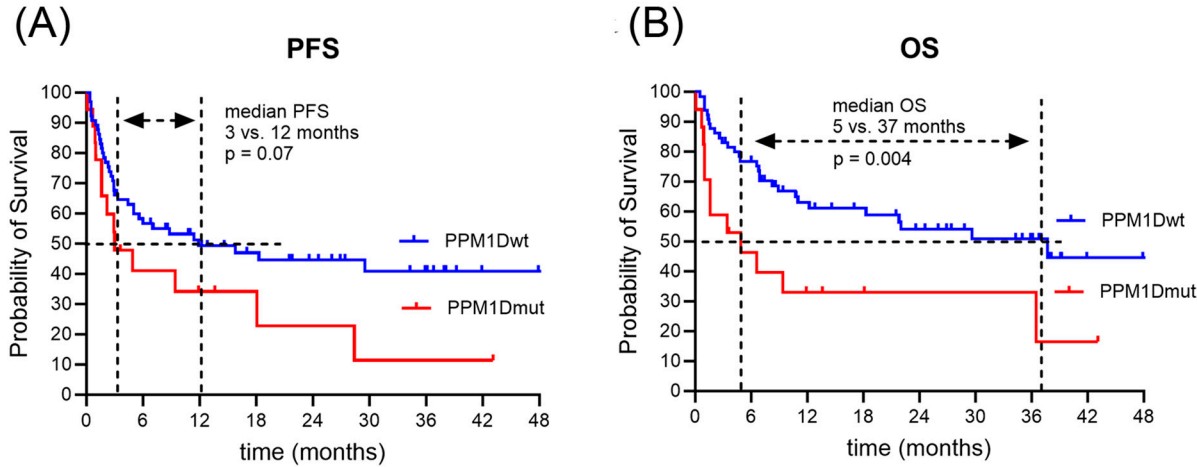

**Figure 3.** Clinical outcome in DLBCL patients treated with CD19-targeted (FMC63) CAR T-cell therapy. PFS (**A**) and OS (**B**) of DLBCL patients receiving CD19-targeted CAR T-cell therapy were analyzed using Kaplan–Meyer, stratified by *PPM1D* status.

The standard approach of applying univariate tests on individual response variables has the advantage of simplicity of interpretation, but fails to account for the covariance and correlation in the data. The multivariate analysis supported the trend observed in the univariate analysis for worse clinical outcomes in DLBCL patients carrying *PPM1D* mutations treated with CAR T therapy, for example, in OS time with a hazard ratio of 2.37 at a *p*-value of 0.016 (Table 5). Similarly, older age at treatment start was associated with worse clinical outcomes in OS time (HR 2.25, *p* = 0.012). Moreover, patients diagnosed with transformed lymphoma had a better treatment outcome than de novo DLBCL, with a hazard ratio of 0.44 (*p* = 0.027), similar to the results previously reported [15]. The serum ferritin levels after CAR T-cell infusion were most significantly associated with treatment outcome.

**Table 5.** Clinical outcome after CAR T-cell therapy, multivariate analysis.

| | PFS | | OS | |
|---|---|---|---|---|
| Predictors | HR (95% CI) | *p*-Value | HR (95% CI) | *p*-Value |
| PPM1Dmut | 1.41 (0.72, 2.74) | 0.3 | 2.37 (1.18, 4.77) | 0.016 |
| Age > 65 | 1.63 (0.92, 2.90) | 0.094 | 2.25 (1.19, 4.26) | 0.012 |
| Sex | 0.81 (0.43, 1.51) | 0.5 | 0.89 (0.44, 1.80) | 0.7 |
| Transformed DLBCL | 0.56 (0.29, 1.06) | 0.074 | 0.44 (0.21, 0.91) | 0.027 |
| Peak ferritin | 1.02 (1.01, 1.03) | <0.001 | 1.03 (1.02, 1.04) | <0.001 |

## 4. Discussion

*PPM1D* is one of the most frequently mutated genes in the context of clonal hematopoiesis (CH). The majority of reported *PPM1D* genomic alterations linked to CH are nonsense or frameshift mutations located within the terminal exon 6 [33,34]. These truncating protein stop-gain or frameshift mutations, which entail a loss of the C-terminal degron domain in Wip1, lead to a gain of function of PPM1D. The resulting hyperactive PPM1D represses the normal p53-dependent G1 checkpoint, thereby promoting cell proliferation [39,40]. In lymphoma patients, mutations in *PPM1D* have been correlated with worse response to standard systemic treatment, as well as shorter OS [30–33,36]. We hypothesized that, by promoting cell cycle dysregulation and cell proliferation, the presence of *PPM1D* mutations may impair optimal response to CAR T-cell therapy.

In our patient cohort, 20% out of the 85 r/r DLBCL patients treated with CAR T-cell therapy harbored mutations within exon 6 of the *PPM1D* gene. This prevalence is consistent with findings from previous publications reporting the prevalence of *PPM1D* somatic mutations ranging between 2 and 23.5% in cancer patients with a history of chemotherapy exposure [30,33,34,45,49]. Moreover, the frequency of *PPM1D* mutations increases with age [33,35]. In our patient cohort, the median age of patients in the PPM1Dmut cohort was 69 years, slightly older than in the PPM1Dwt subgroup, with a median age of 65 years ($p = 0.47$). CH is defined by the presence of cancer-related mutations in hematopoietic cells with a VAF of at least 0.02, in the absence of underlying hematologic malignancy [30,32,34,37]. The mean VAF of 0.052 and median VAF of 0.036 of the *PPM1D* gene variants in our study are reminiscent of the variant allele frequencies detected in CH in response to cytotoxic chemotherapy [45]. Moreover, the identified *PPM1D* exon 6 mutations resembled those previously described [30,32,45], with a majority of stop-gain changes resulting in truncated protein products (15/19, 79%). All PPM1D variants with stop-gain changes are translated into phosphatase protein variants which may not be effectively targeted for degradation by the cellular APC/C complex. Consequently, the function of the p53 tumor suppressor may be impaired in cells with stop-gain mutated *PPM1D* genes, which may lead to enhanced cell survival and proliferation. As the presence of truncated PPM1D proteins leads to inactivation of the tumor suppressor p53, it is possible that *PPM1D* mutations generally associate with adverse outcomes in hematological diseases and other cancers where treatment-induced p53 function is required. To improve treatment outcomes in tumors with *PPM1D* mutations, a selective and potent allosteric PPM1D inhibitor with acceptable pharmacokinetic properties may be applied in the future [50].

The r/r DLBCL patient population analyzed in our study was heavily pre-treated. All carriers of a *PPM1D* mutation had received two or more therapy lines prior to CAR T-cell therapy, in most cases including R-CHOP as standard first-line treatment. Additionally, 63% of the PPM1Dmut population had undergone SCT. Previous studies confirmed an increased prevalence of *PPM1D* mutations in chemotherapy-exposed lymphoma patients. Eskelund et al. reported the expansion of hematopoietic clones carrying mutations in DDR-related genes, such as *PPM1D* or *TP53*, in lymphoma patients after receiving first-line chemotherapy with R-CHOP [51]. Clones with *PPM1D* mutations experienced significantly higher expansion, as compared to clones with other common CH mutations. Moreover, Kahn et al. showed that *PPM1D* mutations were 60 times more likely to be present in chemotherapy-exposed lymphoma patients, even after adjusting for age, as compared to patients without previous malignancy [33]. High-grade CRS and ICANS have been described in DLBCL patients with DTA gene mutations (DNMT3A, TET2, and ASXL1) treated with axi-cel [30]. In our study, there was no significant association between the frequency or severity of CAR T-cell-related adverse events and the presence of *PPM1D* mutations. Nevertheless, there was a tendency for higher-grade ICANS in the PPM1Dmut subgroup ($p = 0.19$). While the majority of ICANS in the PPM1Dwt subgroup was low grade 1 or 2 (56%), ICANS in the PPM1Dmut subgroup tended to be high grade 3 or 4 (75%). Pretreatment serum levels of LDH and inflammatory markers, including CRP, ferritin, and IL-6, however, were elevated in the PPM1Dmut population and negatively

associated with clinical outcomes. Elevated ferritin levels in the sera of DLBCL patients have previously been associated with aggressive disease and poor clinical outcome [52]. Likewise, CRP serum levels >30 mg/L have previously been associated with a shorter OS in DLBCL treated with anti-CD19 CAR T-cell therapy [53].

The outcome of CAR T-cell treatment had previously been analyzed in the same (r,r) DLBCL patient cohort stratified for age, disease status, and CD19 variant [12]: younger patients had a better treatment outcome, with a median OS of 4 years in patients up to 65 years of age and only 7 months in patients over 65 years of age ($p = 0.002$); transformed DLBCL had a better outcome than de novo DLBCL, with a median OS of 8 months for de novo and 3 years for transformed DLBCL ($p = 0.011$); patients carrying the germline variant CD19 L174 allele had a better treatment outcome, with a median PFS of 6 vs. 22 months ($p = 0.06$). In the present study, we detected a significantly worse survival outcome for DLBCL patients with CH-related *PPM1D* mutations (5 versus 37 months; $p = 0.004$). While the prevalent treatment outcome of anti-CD19 CAR T-cell therapy within the PPM1Dwt subgroup was complete remission (56%), the majority of patients within the PPM1Dmut subgroup had only partial remission (60%) and a short time of relapse-free survival (3 versus 12 months; $p = 0.07$). These data suggest a prognostic and possibly predictive impact of *PPM1D* mutations in patients with r/r DLBCL undergoing CD19-targeted CAR T-cell therapy. To confirm the impact of low-frequency *PPM1D* mutations on treatment outcome, a larger retrospective study in DLBCL patients treated with anti-CD19 CAR T-cell therapy is required. Moreover, to improve treatment outcomes in *PPM1D*-mutated DLBCL patients, the administration of a PPM1D inhibitor may present a valid treatment option.

**Author Contributions:** Conceptualization, K.S. and T.P.; investigation, K.S. and M.F.; methodology, K.S.; formal analysis, K.S. and H.N.; resources, Y.B. and U.B.; writing—original draft preparation, M.F. and K.S.; writing—review and editing, K.S., D.A. and T.P.; visualization, K.S.; supervision, K.S. and T.P.; project administration, T.P. All authors have read and agreed to the published version of the manuscript.

**Funding:** This research received no external funding.

**Institutional Review Board Statement:** The study was conducted according to the guidelines of the Declaration of Helsinki, and approved by the Ethics Committee of the Canton of Bern, Switzerland (decision number 2022-00203 and date of approval 4 May 2022).

**Informed Consent Statement:** Informed consent was obtained from all subjects involved in the study according to the Declaration of Helsinki. The general consent form is available.

**Data Availability Statement:** Data are available on request due to restrictions, privacy, and ethics.

**Acknowledgments:** The authors wish to thank the data management, apheresis, flow cytometry, and stem cell laboratory teams of the ASCT program at the University Hospital of Bern and its associated partner hospitals and collaborators for documentation of data relevant to this study.

**Conflicts of Interest:** The authors declare no conflict of interest.

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
