# Peer review of "Low-Frequency PPM1D Gene Mutations Affect Treatment Response to CD19-Targeted CAR T-Cell Therapy in Large B-Cell Lymphoma"

_curroncol, doi:10.3390/curroncol30120762_

Round 1
Reviewer 1 Report
Comments and Suggestions for Authors
General)
The authors described a manuscript entitled “Low frequency PPM1D gene mutations affect treatment response to CD19 targeted CAR T cell therapy in r/r large B-cell lymphoma ” with clinical-practice data of CAR-T cell therapy for Diffuse Large B Cell Lymphoma (DLBCL) in relationship to clonal hematopoiesis (CH) with PPM1D mutation. However, there are several issues to be corrected or clarified.
Majors)
1) The authors analyzed 85 r/r DLBCL patients after CD19 targeted CAR T cell therapy focusing on CH with PPM1D mutation using peripheral blood mononuclear cells and found that the mutation was associated with worse response and poor progression free- and overall- survival but not with toxicities specific to CAR T cell therapy. The authors referred several papers reporting poor prognosis of DLBCL with the CH than without the CH. The authors should discuss the poor prognosis with the CH is specifically associated with CAR T cell therapy or DLBCL in general.
2) Prognostic significance of CH with PPM1D mutation is better to be analyzed with multivariate also.
Minors in “Introduction”)
1; Page 1) “Standard first-line therapy for DLBCL is a combined chemo-immunotherapy with R-CHOP (rituximab, cyclophosphamide, doxorubicin, vincristine and prednisone) [2].”
Polatuzuma vedotin-R-CHP is also a standard first-line therapy for DLBCL.
2; Page 1) “Although R-CHOP is safe and highly effective, 45-50% of patients will relapse [3].”
Progression free survival of DLBCL after R-CHOP is more than 50% in most papers.
3; Page 2) “Patients relapsing after CAR T-cell therapy may be treated with glofitamab [27].”
There are several other Bispecific Monoclonal Antibodies.
Reviewer 2 Report
Comments and Suggestions for Authors
The authors examined the effects of PPM1D gene mutations on the response of patients with large B-cell lymphoma who had received CAR T-cell therapy.
Specific Points of Criticism and Suggestions for Alterations:
(1) Title: Subject-specific abbreviations (clinical slang) should be avoided in the title: better write „resistant/refractory“ (instead of r/r).
(2) Page 2, line 17: Tocilizumab should be shortly explained, for example: " ... includes tocilizumab (a monoclonal antibody against the interleukin-6 receptor) ...".
(3) Page 2, line 23: Also define Glofitamab, for example as „… (a bispecific CD19-CD3 engager monoclonal antibody) …“.
(4) Number of patients studied should be consistent:
- in 2.1. line 2: The cohort comprised 85 patients …“
- in 2.2. line 2: 88 DLBCL patients
- in 3.1. line 2: 88 patients
- in 3.2. line 2: 88 r/r DLBCL
- in Tables 2, 3 and 4: Total cohort (n = 85)
(5) Table 1 could be placed into the supplements.
(6) Figure 2 should be cited in the Introduction and should hence become Figure 1.
(7) Tables 2, 3 and 4: The lists of abbreviations should be in alphabetical order - otherwise, if one looks, for example, for only one specific acronym, one has to scan the whole list.
(8) An Outlook (what to do next? where to go?) would be useful at the end of Discussion.
(9) The limitations of this study should be addressed towards the end of the Discussion.
For example as follows: „The limitations of this study include …“.
Reviewer 3 Report
Comments and Suggestions for Authors
This an interesting and relevant paper that need to be reproduced with larger cohorts and in context with the whole mutational landscape found.
Please read again to find typos. For example
Saini et al reported vs Saini et al.
Thanks
